# Clinical readiness for essential maternal and child health services in Kenya: A cross-sectional survey

Jill M. Hagey[1]*, Sandra Y. Oketch[2,3], Jeremy M. Weber[4], Carl F. Pieper[4], Megan J. Huchko[2,5]

**1** Division of Complex Family Planning, Department of Obstetrics and Gynecology, University of North Carolina-Chapel Hill, Chapel Hill, North Carolina, United States of America, **2** Duke Global Health Institute, Duke University, Durham, North Carolina, United States of America, **3** Kenya Medical Research Institute, Center for Global Health Research, Kisumu, Kenya, **4** Department of Biostatistics and Bioinformatics, Duke University School of Medicine, Durham, North Carolina, United States of America, **5** Department of Obstetrics and Gynecology, Duke University, Durham, North Carolina, United States of America

* jill.hagey@unchealth.unc.edu

## Abstract

High rates of maternal and neonatal morbidity and mortality in Kenya may be influenced by provider training and knowledge in emergency obstetric and neonatal care in addition to availability of supplies necessary for this care. While post-abortion care is a key aspect of life-saving maternal health care, no validated questionnaires have been published on provider clinical knowledge in this arena. Our aim was to determine provider knowledge of maternal-child health (MCH) emergencies (post-abortion care, pre-eclampsia, postpartum hemorrhage, neonatal resuscitation) and determine factors associated with clinical knowledge. Our secondary aim was to pilot a case-based questionnaire on post-abortion care. We conducted a cross-sectional survey of providers at health facilities in western Kenya providing maternity services. Providers estimated facility capacity through perceived availability of both general and specialized supplies. Providers reported training on the MCH topics and completed case-based questions to assess clinical knowledge. Knowledge was compared between topics using a linear mixed model. Multivariable models identified variables associated with scores by topic. 132 providers at 37 facilities were interviewed. All facilities had access to general supplies at least sometime while specialized supplies were available less frequently. While only 56.8% of providers reported training on post-abortion care, more than 80% reported training on pre-eclampsia, postpartum hemorrhage, and neonatal resuscitation. Providers' clinical knowledge across all topics was low (mean score of 63.3%), with significant differences in scores by topic area. Despite less formal training in the subject area, providers answered 71.6% (SD 16.7%) questions correctly on post-abortion care. Gaps in supply availability, training, and clinical knowledge on MCH emergencies exist. Increasing training on MCH topics may decrease pregnancy and postpartum complications. Further, validated tools to assess knowledge in post-abortion care should be created, particularly in sub-Saharan Africa where legal restrictions on abortion services exist and many abortions are performed in unsafe settings.

**Data Availability Statement:** The data that support the findings of this study are openly available in the Duke Digital Repository for Research Data at http://doi.org/10.7924/r47m0fd7k.

**Funding:** This work was supported by the Charles B. Hammond Research Fund of Duke Obstetrics and Gynecology (Grant #2910215 to JMH). Salary support was provided to JMW and CFP from the National Center for Advancing Translational Sciences (NCATS) of the National Institutes of Health (Grant #UL1T002553 to JMW and CFP). The contents of the manuscript are solely the responsibility of the authors and do not necessarily represent the official views of NCATS or NIH. The funder had no role in study design, data collection and analysis, decision to publish, or preparation of the manuscript.

**Competing interests:** The authors have declared that no competing interests exist.

## Introduction

Pregnancy at the extremes of age (both increased teen pregnancy and pregnancy at advanced maternal age) and increasing rates of non-communicable diseases (NCDs) among pregnant women have elevated the risk of morbidity and mortality during pregnancy and childbirth in low- and middle-income countries (LMIC). LMIC also struggle to respond to obstetric-related emergencies, which are more likely in high-risk pregnancies. Extremes in age of motherhood (both younger and older) are associated with hypertensive diseases, gestational diabetes, fetal macrosomia, miscarriage, and stillbirth [1]. Similarly, diseases such as chronic hypertension and pre-gestational diabetes increase the risk of pre-eclampsia, postpartum hemorrhage, pre-term delivery, fetal growth restriction, and miscarriage or stillbirth [2, 3]. Globally, these diseases have resulted in significant maternal mortality, with postpartum hemorrhage (27.1%), hypertensive disorders (14.0%), and sepsis (10.7%) causing over half of maternal deaths worldwide [4]. Targeting these complications may be necessary to prevent maternal deaths and achieve the Sustainable Development Goal of a maternal mortality ratio lower than 70 per 100,000 live births by 2030 [5]. Kenya's maternal mortality ratio remains high at 530 per 100,000 live births and has been increasing in recent years [6].

In 2011, the World Health Organization (WHO) Partnership for Maternal, Newborn and Child Health set out a list of essential interventions in maternal-child health (MCH), including post-abortion care, management of pre-eclampsia and eclampsia, management of postpartum hemorrhage, and neonatal resuscitation [7]. The goal to provide essential MCH services aligns with broader recommendations to ensure that emergency obstetric care is available to all women and newborns [8]. Kenya, along with other LMIC, abolished delivery fees in all public health facilities in 2013 to increase the number of births attended by a skilled provider able to respond to these emergencies [9].

Availability of appropriate MCH interventions in the face of maternal and neonatal emergencies is dependent on the number and training of health care providers along with measures of facility readiness to provide care (facility space, supplies and equipment). However, despite a global focus on essential MCH care, access to appropriate MCH interventions is often lacking in LMIC. In a cross-sectional study of six countries including Kenya, only 2.3% of surveyed health facilities had the supplies and tools expected for basic emergency obstetric care [10]. Further, recent data suggests that monitoring and reporting supply availability alone for MCH care often overestimates health facilities' ability to provide these services [11, 12]. Indeed, even in facilities with appropriate infrastructure for MCH interventions, there is still significant heterogeneity in adherence to best clinical practices [13]. Information on training and provider knowledge must be utilized in conjunction with facility assessments of supply availability for a comprehensive assessment of competent facilities in MCH interventions.

Subject-specific knowledge questionnaires in MCH topics have been used in multiple contexts in sub-Saharan Africa to evaluate the success of training modules and integration of skills into daily activities [14, 15]. When case-based questionnaires were used to assess knowledge acquisition following team training and simulation learning on a variety of MCH emergencies, improvements in clinical knowledge were seen from pre-training to post-training [14, 15]. While clinical vignettes exist to evaluate many MCH topics, knowledge evaluations on post-abortion care are lacking. Post-abortion care is a key component of MCH services; however, provider knowledge questionnaires often focus on the legality around post-abortion services instead of on care provision based on international standards [16–19]. Indeed, only one prior assessment on provider knowledge of post-abortion care in LMIC focused on treatment and potential complications [20].

To ensure that women receive the quality care during pregnancy and delivery that will improve maternal morbidity and mortality, it is necessary to understand not only facility availability of supplies and services but also training and clinical knowledge of healthcare professionals. Additional focus on these interventions around delivery in Kenya will be essential in accelerating reductions in mortality to help reach the Sustainable Development Goals by 2030 [21]. While prior studies have increased the specificity of measuring clinical readiness through identifying resources to identify, treat, and monitor or modify MCH emergencies at different health facilities; they lack information on whether providers have been trained to use these supplies or have current knowledge on how to use these supplies [11, 12]. The aim of this study was further improve the accuracy in the diagnosis of clinical readiness by measuring not just supply availability, but also an assessment of provider training and knowledge. We sought to do this across four MCH topics: post-abortion care, management of pre-eclampsia and eclampsia, management of postpartum hemorrhage and neonatal resuscitation. Our secondary aim was to trial a case-based questionnaire on post-abortion care given the gaps in the literature.

## Materials and methods

### Ethics statement

The study received ethics approval from Great Lakes University of Kisumu Research Ethics Committee (ID GREC 019/19) for human subjects' research. The study received ethics exemption from Duke Health Institutional Review Board (Pro00100344) for research involving the use of educational tests, survey procedures, interview procedures or observation of public behavior given that the information was collected in a non-identifiable manner. All study methods were performed in accordance with relevant guidelines and regulations.

### Data collection

We conducted a cross-sectional survey among a convenience sample of maternity providers involved in inpatient MCH care provision in western Kenya (S1 Table). Study personnel identified prospective participants with the help of local research assistants, confirmed inclusion criteria, and obtained informed consent. Participants were included in the study if they were a health care provider at a Ministry of Health supported health facility at the sub-county or county hospital level (Kenya Essential Package of Health [KEPH] level 4 or 5) in the surveyed counties [22]; provided inpatient MCH care services; completed medical training at nursing level or higher (nurse/midwife, nursing officer, clinical officer, medical officer); and were able to provide written informed consent. Sub-county and county hospitals (KEPH levels 4 and 5) were chosen for inclusion as these facilities are the minimum tier which respond to complications during pregnancy and obstetric emergencies [23]. Participants were excluded if they could not provide informed consent or respond to the survey questionnaire in English.

Participants completed verbally administered interviews with study personnel. The survey included questions regarding facility demographics, provider demographics, clinical knowledge regarding the four MCH topics, facility training, and available supplies at the facility. Providers were read survey questions by members of the study team in a private location at each of the study health facilities. Answers were recorded on either paper surveys or handheld tablets and later uploaded to a centralized secure Qualtrics database (SAP, Walldorf, Germany), which was then transferred to a secure server to use for data analysis. Supplies assessed at each facility were based on specific commodities from the Demographic Health Survey Service Provision Assessment and clinical knowledge was based on assessments of provider trainings from a non-profit organization (PRONTO) instructing in these areas [15, 24–26]. Knowledge

questionnaires from PRONTO were developed by the study team and reviewed by experts prior to their use in assessing clinical knowledge among health care providers. No formal validation process was followed, but these knowledge questionnaires have been tested across multiple low- and middle-income countries worldwide [14, 15, 26, 27]. General supplies and specific supplies for each intervention were selected from the Kenya Service Provision Assessment and studies investigating the care provision for maternal and neonatal emergency interventions [11, 12, 24]. Six questions each in three of the four subject matters (management of pre-eclampsia and eclampsia, management of postpartum hemorrhage and neonatal resuscitation) were adapted from prior PRONTO trainings covering broad areas of diagnosis, treatment and monitoring of complications associated with these topics. As no prior validated case-based assessments on post-abortion care had been conducted in clinical providers, we designed knowledge questions for this topic area. Questions from the other topic areas were used as a guide to create a similar scope and level of difficulty of the questions. These questions were reviewed by the study team (JH, MH) prior to use in the study survey.

## Outcome measures

Participant and facility demographics were summarized with descriptive statistics. The availability of training by MCH topic and perceived supply availability were reported at the participant level.

Supply availability data was also aggregated for descriptive analyses given that it was reported by each participant. For each participant and MCH topic, the percentage of supplies that were never, rarely, sometimes, or always available were calculated. To understand supply availability between facility location and level, the Likert scale of never to always available was converted to a numeric scale of 0 to 3 such that higher scores indicate higher availability.

Clinical knowledge was assessed by the proportion of correct responses to the set of multiple-choice questions in each of the four MCH topics.

## Statistical analysis

Continuous variables were summarized using mean (standard deviation; SD) or median (Q1 = 25th percentile, Q3 = 75th percentile), and categorical variables were described as frequency and percentage. Likert scale questions regarding supply availability were averaged across the supplies in each MCH topic within a provider and summarized by facility location and level using median (Q1, Q3). Variability in the availability of each type or category of supplies was divided into within facility and between facility variability and reported as an intraclass correlation coefficient (ICC) to assess differences in supply availability between facilities.

Mean composite scores for all clinical knowledge multiple-choice questions between health professional levels and facility levels were compared by t-tests. Linear mixed effects models with random intercepts for participant and facility, to account for responses being clustered within participants and participants being clustered within facilities, were fit to compare the proportion of correct questions between MCH topics with post-abortion care chosen as the reference. Due to the limited sample size four separate models were fit adjusting for different groups of covariates: (1) an unadjusted model, (2) adjusting for demographics, (3) adjusting for training, and (4) adjusting for supply availability. Marginal models using generalized estimating equations (GEE) were fit to identify provider- and facility-level factors that are associated with the percentage of correct questions. Similarly, a GEE-type model was chosen to account for responses being clustered within participants. Mean differences (betas) from the regression models, 95% confidence intervals (CIs), and p-values were reported. The linearity assumption for continuous covariates was assessed using a lack-of-fit test comparing the linear

fit to a restricted cubic spline fit with knots placed at the $10^{th}$, $50^{th}$, and $90^{th}$ percentiles. If a significant non-linear relationship was observed, then a plot with the restricted cubic spline fit was examined to determine an appropriate knot for linear splines.

Factor analysis was performed to investigate if any of the post-abortion care questions measured certain common aspects of clinical knowledge. Latent class analysis was also performed on the post-abortion care questions to examine if clusters of providers could be identified. As this is the first trial of these post-abortion care case vignettes, these analyses were exploratory only. Rigorous validation of this instrument was beyond the scope of this study.

All analyses were completed in SAS 9.4 (SAS Institute, Cary, NC) at a significance level of 0.05, two-tailed. As these analyses were exploratory, significance tests were performed without p-value adjustment.

## Results

### Participant and facility demographics

One hundred and thirty-two participants were interviewed from 37 health facility sites in Homa Bay, Kisumu, and Migori counties. One additional health professional was approached for inclusion but declined participation in the study for a survey response rate of 99%. Most participants worked at a sub-county hospital (116 participants; 87.9%), and the mean (SD) age was 36 (8) years. Nearly 60% of participants were female and over 80% had nursing training. Participants worked a median (Q1, Q3) of 7 (4, 14) years in their roles and were a median (Q1, Q3) of 9 (5, 16) years out from training. Most participants had completed clinical training in Kenya, though most did not train in the county where they were currently working (Table 1).

### Training

Many providers had received training in the four MCH topics (Table 2). Participants were least likely to be trained in post-abortion care at 56.8%. Most participants reported having other health facility members trained across other MCH topics even if they themselves were not trained. However, participants who reported they were not trained in a specific MCH topic were less likely to indicate there were other health facility members trained in that topic compared to when the index participant was trained in the MCH topic. Self-reported numbers of colleagues trained was lowest in post-abortion care compared to other MCH topic areas.

### Resources

Among specific MCH resources, over 80% of supplies for managing pre-eclampsia and eclampsia, post-abortion care and neonatal resuscitation were reported as sometimes or always available (Table 3). However, only 56.2% of supplies for managing postpartum hemorrhage were reported as sometimes or always available.

Supply availability was similar across the different counties sampled. Most sub-county hospitals had similar supply availability to county hospitals except for postpartum hemorrhage supplies, which were perceived as more likely to be available at county hospitals (Table 4). This may be due to lack of blood products (43.8% available at county vs. 17.2% at sub-county hospitals) and second line uterotonics (62.5% available at county vs. 8.6% at sub-county hospitals).

Intraclass correlation coefficients (ICCs) were used to assess if perceived supply availability varied between hospitals (S2 Table). Many general supplies and crucial supplies for specific MCH interventions (including those previously used as signal functions to indicate facilities readiness) had low ICCs, indicative of uniform availability. Differences between facilities (high ICCs) were seen among postpartum hemorrhage supplies and newborn resuscitation tables.

**Table 1. Participant and facility demographics (N = 132).**

| Characteristic | Value |
|---|---|
| **Facility Location, N (%)** | |
| Homa Bay County | 41 (31.1%) |
| Kisumu County | 57 (43.2%) |
| Migori County | 34 (25.8%) |
| **Facility Type, N (%)** | |
| Sub-County Hospital (KEPH Level 4) | 116 (87.9%) |
| County Hospital (KEPH Level 5) | 16 (12.1%) |
| **Provider age, mean (SD)** | 36.4 (8.1) |
| **Provider sex, N (%)** | |
| Male | 55 (41.7%) |
| Female | 77 (58.3%) |
| **Provider role, N (%) [a]** | |
| Medical Officer | 4 (3.0%) |
| Clinical Officer | 20 (15.2%) |
| Nursing Officer/Midwife/Registered Nurse | 108 (81.8%) |
| **Years in role, median (Q1, Q3)** | 7 (4, 14) |
| **Years since initial training, median (Q1, Q3)** | 9 (5, 16) |
| **Training Location, N (%)** | |
| Local (Kisumu, Homa Bay, Migori county) | 47 (35.6%) |
| National (elsewhere in Kenya outside of Homa Bay, Kisumu and Migori counties) | 75 (56.8%) |
| International | 10 (7.6%) |

[a] Provider roles include medical officers (medical doctors with required internship and often specialty training), clinical officers (medical doctors with a shorter training program and often not specialized), nursing officers and midwives (advanced practice providers) and registered nurses.

## Clinical knowledge

Participants answered on average 63.3% (SD 11.8%) of the clinical knowledge questions correctly across all four MCH topics. Mean composite scores were not statistically different by facility level or provider role (mean percentage correct 63.7% among medical and clinical officers versus 63.2% among nurses, p = 0.87; mean percentage correct 63.4% in sub-county hospitals versus 62.8% in county hospitals, p = 0.87). Training in an increasing number of MCH topic areas (regardless of specific training) was associated with increased clinical knowledge. Training in one, two, three, or four MCH topics was associated with a 6.0% (95% CI = -0.09, 12.2), 8.3% (95% CI = 1.9, 14.8), 8.4% (95% CI = 2.5, 14.3), and 11.2% (95% CI = 5.2, 17.2) increase in the percentage of correct questions across all topics tested (p = 0.03), respectively. Age was also associated with clinical knowledge (p = 0.035), but a non-linear relationship was

**Table 2. Participant training on obstetric and neonatal interventions.**

| Training Characteristic | Post-Abortion Care | Pre-eclampsia/ Eclampsia | Postpartum Hemorrhage | Neonatal Resuscitation |
|---|---|---|---|---|
| **Participant Trained, N (%)** | 75 (56.8%) | 113 (85.6%) | 110 (83.3%) | 109 (82.6%) |
| **Other Members of Health Facility Trained, N (%)** | 106 (80.3%) | 115 (87.1%) | 115 (87.1%) | 110 (83.3%) |
| **Other Members of Health Facility Trained if Participant Not Trained, N (%)** | 43 (75.4%) | 11 (57.9%) | 13 (59.1%) | 14 (60.9%) |
| **Number of Health Facility Members Trained, median (Q1, Q3)** | 3 (1, 6) | 5 (2, 10) | 6 (3, 10) | 4.5 (1, 10) |

**Table 3. Participant perceived supply availability for obstetric and neonatal interventions.**

| Supply Availability (%) [a] | General Supplies | Post-Abortion Care | Pre-eclampsia/ Eclampsia | Postpartum Hemorrhage | Neonatal Resuscitation |
|---|---|---|---|---|---|
| Always Available | 79.8 (17.7) | 66.9 (20.4) | 55.3 (28.6) | 42.7 (18.3) | 71.4 (18.6) |
| Sometimes Available | 16.7 (16.4) | 16.9 (17.0) | 30.6 (27.0) | 13.5 (16.1) | 12.1 (15.4) |
| Rarely Available | 2.1 (6.9) | 3.3 (7.5) | 7.7 (16.7) | 5.0 (9.4) | 2.8 (6.8) |
| Never Available | 1.0 (3.8) | 11.1 (13.1) | 5.8 (13.4) | 37.4 (20.7) | 13.2 (13.2) |

[a] All numbers presented as mean (SD)

observed. Linear splines with a knot at 40 years were created to account for the non-linearity. When age was less than or equal to 40 years, a five-year increase was associated with a 2.6% increase in the percentage of questions across all topics. When age was greater than 40 years, a five-year increase was associated with a 3.1% decrease in the percentage of correct questions. Supply availability, facility level, facility county, professional level, number of years in role, number of years since training, and sex were not associated with increased or decreased knowledge of individual MCH interventions or overall knowledge.

Health care providers had statistically significant differences in clinical knowledge for individual MCH topics in both unadjusted and adjusted models. The providers tended to score highest on the post-abortion care questions with a mean (SD) score of 71.6% (16.7%), followed by postpartum hemorrhage at 69.6% (21.2%), neonatal resuscitation at 56.8% (19.1%), and lastly pre-eclampsia and eclampsia at 55.3% (20.5%) (Table 5). These differences in knowledge were statistically significant in the unadjusted model and adjusted models (p <0.0001 for all). Estimates and p-values were the same for the unadjusted model and all three adjusted models: one adjusting for demographics (facility county and level, provider role, age, sex, years in role); one adjusting for training (number of MCH topics ever trained on, number of MCH topics trained on within the last 3 years, years since completing medical training); and one adjusting for supply availability. Comparing to postpartum hemorrhage, pre-eclampsia/eclampsia and neonatal resuscitation scores were still significantly lower (p < 0.001 for both comparisons). Marginal models with generalized estimating equations did not identify an association of facility or provider level factors with knowledge of individual MCH interventions (S3 Table).

As the case-based questions for post-abortion care were newly created for this study, a factor analysis was conducted to determine which questions were most useful in assessing knowledge in this topic area. No specific questions showed a statistically significant association with overall clinical knowledge score on the post-abortion care questions. A latent class analysis separated the participants into 3 classes of performers on the post-abortion care questions.

**Table 4. Participant perceived supply availability for obstetric and neonatal interventions, by facility location and level.**

| Average Supplies Available [a] | Post-Abortion Care | Pre-eclampsia/ Eclampsia | Postpartum Hemorrhage | Neonatal Resuscitation |
|---|---|---|---|---|
| **Facility Location** | | | | |
| Kisumu County | 2.5 (2.3, 2.8) | 2.6 (2.2, 2.8) | 1.6 (1.4, 2.2) | 2.6 (2.3, 2.8) |
| Homa Bay County | 2.3 (2.0, 2.6) | 2.2 (1.8, 2.6) | 1.2 (1.2, 1.6) | 2.4 (2.0, 2.6) |
| Migori County | 2.5 (2.2, 2.8) | 2.4 (2.0, 2.8) | 1.9 (1.4, 2.2) | 2.4 (2.1, 2.8) |
| **Facility Level** | | | | |
| Sub-County Hospital (KEPH Level 4) | 2.4 (2.2, 2.8) | 2.4 (2.0, 2.8) | 1.4 (1.2, 2.0) | 2.4 (2.1, 2.6) |
| County Hospital (KEPH Level 5) | 2.6 (2.4, 2.9) | 2.6 (2.3, 3.0) | 2.2 (2.0, 2.4) | 2.7 (2.4, 2.9) |

[a] All numbers presented as median (Q1, Q3); scores range from 0 (never available) to 3 (always available) such that higher numbers indicated higher availability.

**Table 5. Clinical knowledge on obstetric and neonatal interventions.**

| Clinical Knowledge | Post-Abortion Care | Pre-eclampsia/ Eclampsia | Postpartum Hemorrhage | Neonatal Resuscitation | Total Survey |
|---|---|---|---|---|---|
| Percentage of Clinical Knowledge Questions Correct, mean (SD) | 71.6 (16.7) | 55.3 (20.5) | 69.6 (21.2) | 56.8 (19.1) | 63.3 (11.8) |
| Unadjusted Mean Difference (95% CI) [a] | Reference | -16.3 (-20.6, -12.0); p <0.0001 | -2.0 (-6.3, 2.3); p = 0.36 | -14.8 (-19.1, -10.5); p <0.0001 | N/A |

[a] Adjusted models by demographic factors, training factors, and supply availability factors produced the same estimates and p-values as the unadjusted model.

The worst performers (N = 7, 5.3%) scored a median of two out of six questions correct, the middle performers (N = 24, 18.2%) scored a median of four out of six questions correct, and the best performers (N = 101, 76.5%) scored a median of five out of six questions correct. The worst performing group size was too small to determine any facility or provider characteristics associated with poor performance (Table 6).

Finally, clinical knowledge questions across all four MCH topics were separated by question type (asking about diagnosis, treatment, or monitoring obstetric or neonatal emergency) and correlated with provider and facility factors. Male providers scored on average 8.9% higher on questions regarding how to monitor emergencies (66.1% versus 57.1%; 95% CI = 2.3, 15.6; p = 0.009). Age was also significantly associated with knowledge on how to monitor emergencies (p = 0.037). The relationship was non-linear so linear splines were created with a knot at 35 years. There was not a statistically significant association when age was less than or equal to 35 years. However, beyond 35 years, a five-year increase was associated with a 4.2% reduction in the percentage of correct questions.

**Table 6. Participant demographics based on performance on post-abortion care questions.**

| | Worst performers (N = 7) | Middle performers (N = 24) | Best performers (N = 101) | Total (N = 132) |
|---|---|---|---|---|
| **Facility Location, N (%)** | | | | |
| Homa Bay County | 3 (42.9%) | 7 (29.2%) | 31 (30.7%) | 41 (31.1%) |
| Kisumu County | 2 (28.6%) | 11 (45.8%) | 44 (43.6%) | 57 (43.2%) |
| Migori County | 2 (28.6%) | 6 (25.0%) | 26 (25.7%) | 34 (25.8%) |
| **Facility Type, N (%)** | | | | |
| County Hospital (KEPH Level 5) | 1 (14.3%) | 4 (16.7%) | 11 (10.9%) | 16 (12.1%) |
| Sub-County Hospital (KEPH Level 4) | 6 (85.7%) | 20 (83.3%) | 90 (89.1%) | 116 (87.9%) |
| **Provider age, mean (SD)** | 40.0 (9.2) | 34.4 (7.6) | 36.7 (8.1) | 36.4 (8.1) |
| **Provider sex, N (%)** | | | | |
| Male | 2 (28.6%) | 10 (41.7%) | 43 (42.6%) | 55 (41.7%) |
| Female | 5 (71.4%) | 14 (58.3%) | 58 (57.4%) | 77 (58.3%) |
| **Provider role, N (%)** | | | | |
| Medical Officer | 0 (0.0%) | 3 (12.5%) | 1 (1.0%) | 4 (3.0%) |
| Clinical Officer | 0 (0.0%) | 6 (25.0%) | 14 (13.9%) | 20 (15.2%) |
| Nursing Officer/Midwife/Registered Nurse | 7 (100.0%) | 15 (62.5%) | 86 (85.1%) | 108 (81.8%) |
| **Years in role, median (Q1, Q3)** | 10 (5, 14) | 6 (3, 11) | 7 (4, 15) | 7 (4, 14) |
| **Years since training, median (Q1, Q3)** | 11 (5, 16) | 7.5 (3.5, 12.5) | 9 (5, 18) | 9 (5, 16) |
| **Training location, N (%)** | | | | |
| Local | 2 (28.6%) | 5 (20.8%) | 40 (39.6%) | 47 (35.6%) |
| National | 4 (57.1%) | 17 (70.8%) | 54 (53.5%) | 75 (56.8%) |
| International | 1 (14.3%) | 2 (8.3%) | 7 (6.9%) | 10 (7.6%) |
| **Subspecialty trained, N (%)** | 5 (71.4%) | 8 (33.3%) | 37 (36.6%) | 50 (37.9%) |

## Discussion

Facility and health care provider readiness to respond to MCH emergencies are key in reducing maternal mortality. We assessed the clinical knowledge, training, and commodities necessary for management of four key MCH areas (post-abortion care, pre-eclampsia and eclampsia, postpartum hemorrhage, and neonatal resuscitation) among inpatient maternity units in Homa Bay, Kisumu and Migori counties in western Kenya. While most inpatient maternity wards surveyed had access to general supplies and training in MCH topic areas, we identified a lack of provider knowledge that may impact provision of quality obstetrics and neonatal care.

General commodities for MCH interventions were often available in the facilities surveyed, with 80% of participants reporting them as always available. Availability of oxytocin, misoprostol, and magnesium sulfate were higher in our sample than reported for western Kenya in prior surveys from 2013 [27]. However, more specialized, and often expensive, supplies were less well stocked. Supplies to manage postpartum hemorrhage, including ergometrine and blood transfusion supplies, were often noted as never available, nor were temporizing devices such as non-pneumatic shock garments. We also noted that the availability of IV fluids, antibiotics, uterotonics and manual vacuum aspirators for removal of retained products of conception was low, which is comparable with prior studies on health facility capacity in Kenya and other sub-Saharan African countries to provide post-abortion care [28].

Over 80% of participants were trained in pre-eclampsia/eclampsia, postpartum hemorrhage, and neonatal resuscitation. Among those not trained in specific MCH topics, the majority reported at least one colleague trained in the topic area. This is important as these providers can rely on colleagues to support them if a clinical scenario is out of their comfort level. However, those not trained in specific topic areas were less likely to report others in their facility trained in those areas, suggesting that lack of training may be localized within specific clinics. This may be a true lack of training at these sites or perceived lack of training by those not trained in these areas. Prior studies have indicated that trainings, and specifically simulation-based training, in MCH topics has resulted in improved knowledge on treating MCH emergencies in a variety of LMIC settings [29].

Training in post-abortion care was less common. This may be due to a temporary halt in documentation of standards and guidelines on how to provide appropriate post-abortion care services following changes to the Kenyan Constitution in 2010, which legalized abortion only when the life or health of a woman are in danger [30]. Many providers did not know how to interpret these legal changes in their scope of practice, reducing the amount of formal training in this area [31]. Training is an important avenue for continued attention, given that training across any of the MCH topics was associated with improved clinical knowledge across all domains tested in the survey. Interestingly, provider knowledge on post-abortion care was higher than the other three MCH topic areas despite the lowest amount of training, which may be indicative of informal or on-the-job training rather than specific training programs. However, there is likely a role for refresher courses or additional training, as those individuals that were among the worst performers on the post-abortion care vignettes tended to be further out from their formal schooling. Factor analysis of the post-abortion care vignettes did not yield any additional information on questions that may have been the most sensitive to assessing provider knowledge in this domain, and further study is still needed to validate a questionnaire to assess knowledge in this area.

Providers correctly answered over 60% of the clinical knowledge questions across the four health topics correctly, highlighting the gap between training and knowledge. There were no significant differences in either overall or specific MCH topic knowledge based on provider or

facility factors, however clinical knowledge was lowest with regards to pre-eclampsia. These low scores are concordant with other literature reporting provider knowledge and subsequent appropriate treatment of pre-eclampsia and eclampsia among 6 countries in sub-Saharan Africa [32]. Many providers attributed lack of knowledge to overall low rates of pre-eclampsia at their facilities; however, as the prevalence of hypertension rises, the rates of pre-eclampsia are likely to rise in the coming years. While some maternal and neonatal complications may be rare at individual facilities, training in emergency obstetric care addresses the main causes of maternal and neonatal death and has been associated with improved health outcomes world-wide [33].

This study had many strengths. By sampling providers across multiple health facility levels in geographically disparate areas, we increased our generalizability to other LMIC settings. Additionally, we used previously tested questions regarding supply availability, training, and clinical knowledge to help strengthen the reproducibility of these results. There were also some limitations, including the use of convenience sampling. However, only one provider declined participation in the study and we exceeded our proposed number of participants and obtained large enough numbers across all demographic groups to make relevant comparisons. Another limitation is that participants self-reported both their perception of supply availability and training completed by other colleagues in their health centers, potentially biasing these results with either over-or under-reporting. However, our data is comparable to similar studies conducted in analogous settings, and we used the clinical knowledge data as a more objective measure. Some of the differences in post-abortion care knowledge may be due to potential different difficulty level of these questions compared to the other clinical knowledge questions. Finally, given the number of factors we investigated as potentially related to clinical knowledge, we did risk overstating statistical significance due to multiple comparisons.

## Conclusions

By evaluating clinical knowledge, training, and supply availability across a broad spectrum of MCH topics in western Kenya, we were able to develop a more holistic understanding of MCH care provision. Further studies could use these aspects of readiness to build a composite score that can be utilized across different facilities. A multidisciplinary approach to ensure adequate provider knowledge and address supply chain gaps is necessary to decrease complications of pregnancy and childbirth in western Kenya. Attention to rigorous training programs, especially simulation-based training, can help improve provider knowledge; and care should be made to focus on improving access to supplies needed for basic EmOC interventions. Specific attention is needed to increase training on post-abortion care and validate measures to assess clinical knowledge in this domain.

## Supporting information

**S1 Table. Facilities in Homa-Bay, Kisumu and Migori counties included in study.**
(DOCX)

**S2 Table. Provider perception of supply availability by obstetric and neonatal interventions and intraclass correlation coefficient (ICC).**
(DOCX)

**S3 Table. Provider and facility level factor associations with individual MCH topics by generalized estimating equations.**
(DOCX)

## Acknowledgments

The authors would like to acknowledge the following research assistants for their help in data collection for the study: Faith Otewa, Breandan Makhulo, Kenneth Otieno, Evans Otieno, Belinda Malongo, Suzanna Larkin, and Emma Mehlhop. The team would also like to thank the County Directors for Health from Homa Bay County (Dr. Gordon Okomo), Kisumu County (Dr. Dickens Onyango) and Migori County (Dr. Elizabeth Mgamb).

## Author Contributions

**Conceptualization:** Jill M. Hagey, Megan J. Huchko.

**Data curation:** Jeremy M. Weber.

**Formal analysis:** Jeremy M. Weber, Carl F. Pieper.

**Funding acquisition:** Jill M. Hagey, Megan J. Huchko.

**Investigation:** Jill M. Hagey.

**Methodology:** Jill M. Hagey, Sandra Y. Oketch, Megan J. Huchko.

**Project administration:** Jill M. Hagey, Sandra Y. Oketch.

**Resources:** Megan J. Huchko.

**Supervision:** Carl F. Pieper, Megan J. Huchko.

**Writing – original draft:** Jill M. Hagey.

**Writing – review & editing:** Jill M. Hagey, Sandra Y. Oketch, Jeremy M. Weber, Carl F. Pieper, Megan J. Huchko.

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
