## [Decision Letter · Decision Letter 0]

11 Apr 2023

PGPH-D-22-01510

Clinical knowledge among health care providers of essential maternal and child health services in Kenya: A cross-sectional survey

Dear Dr. Hagey,

Thank you for submitting your manuscript to PLOS Global Public Health. After careful consideration, we feel that it has merit but does not fully meet PLOS Global Public Health’s publication criteria as it currently stands. Therefore, we invite you to submit a revised version of the manuscript that addresses the points raised during the review process.

Please be sure to address the reviewers' comments, especially issues of biases at various stages of data collection. In addition to the comments below you will find additional comments as an attachment.

We look forward to receiving your revised manuscript.

Kind regards,

Adriana Andrea Ewurabena Biney

Academic Editor

Journal Requirements:

1. Please ensure that Funding Information and Financial Disclosure Statement are matched.

Additional Editor Comments (if provided):

Reviewers' comments:

Reviewer's Responses to Questions

**Comments to the Author**

1. Does this manuscript meet PLOS Global Public Health’s publication criteria? Is the manuscript technically sound, and do the data support the conclusions? The manuscript must describe methodologically and ethically rigorous research with conclusions that are appropriately drawn based on the data presented.

Reviewer #1: Yes

Reviewer #2: Partly

2. Has the statistical analysis been performed appropriately and rigorously?

Reviewer #1: Yes

Reviewer #2: Yes

3. Have the authors made all data underlying the findings in their manuscript fully available (please refer to the Data Availability Statement at the start of the manuscript PDF file)?

Reviewer #1: Yes

Reviewer #2: Yes

4. Is the manuscript presented in an intelligible fashion and written in standard English?

Reviewer #1: Yes

Reviewer #2: Yes

5. Review Comments to the Author

Reviewer #1: Dear Author

Thanks for your submission

It is an interesting paper but needs some work on clarity

I think you need to make it very clear throughout the abstract, introduction and methods that you are looking at overall knowledge re medical emergencies and then the knowledge of each of them

While you trialled a questionnaire about post abortion care as you did not test validity, reliability etc of this I would just state that you needed to design your own methods to collect this data ( and address limitations of this in limitation section of discussion) and then focus on describing knowledge etc across all and each medical emergencies

Otherwise the reader is confused as to whether this is a paper on postabortion care in which case a different introduction is required or medical emergencies for MCH ( I would go with the latter)

In the Intro line 77 could you please give some information on what were the results from 12 and 13. I would also suggest considering a strengths based where possible approach to reporting results and where are clinicians doing well ?

In the methods please use headings and separate out outcome measures from analysis as it is currently mixed together and confusing

Please give a reference and short sentences on the validation of PRONTO training questions

Results

I am confused why 95% confidence intervals are not used when reporting the outcomes in Table 3?

I would like to see clearer recommendations for services and research as a result of your findings that are feasible in a LMIC setting

Reviewer #2: The manuscript is an interesting read, however the title is not quite clear. The body of the manuscript carries a lot of observations that are not related to the title of study. It requires modification of the title to make it as comprehensive as much as possible comprehensive

6. PLOS authors have the option to publish the peer review history of their article (what does this mean?). If published, this will include your full peer review and any attached files.

**Do you want your identity to be public for this peer review?** For information about this choice, including consent withdrawal, please see our Privacy Policy.

Reviewer #1: No

Reviewer #2: **Yes: **Dr. Angeline C. Kirui, RN, PhD

---

## [Decision Letter · Decision Letter 1]

19 Jul 2023

PGPH-D-22-01510R1

Clinical readiness for essential maternal and child health services in Kenya: A cross-sectional survey

Dear Dr. Hagey,

Thank you for submitting your manuscript to PLOS Global Public Health. After careful consideration, we feel that it has merit but does not fully meet PLOS Global Public Health’s publication criteria as it currently stands. Therefore, we invite you to submit a revised version of the manuscript that addresses the points raised during the review process.

We look forward to receiving your revised manuscript.

Kind regards,

Adriana A. E. Biney

Academic Editor

Journal Requirements:

Additional Editor Comments (if provided):

Reviewers' comments:

Reviewer's Responses to Questions

**Comments to the Author**

1. If the authors have adequately addressed your comments raised in a previous round of review and you feel that this manuscript is now acceptable for publication, you may indicate that here to bypass the “Comments to the Author” section, enter your conflict of interest statement in the “Confidential to Editor” section, and submit your "Accept" recommendation.

Reviewer #2: All comments have been addressed

Reviewer #3: (No Response)

2. Does this manuscript meet PLOS Global Public Health’s publication criteria? Is the manuscript technically sound, and do the data support the conclusions? The manuscript must describe methodologically and ethically rigorous research with conclusions that are appropriately drawn based on the data presented.

Reviewer #2: Yes

Reviewer #3: Partly

3. Has the statistical analysis been performed appropriately and rigorously?

Reviewer #2: Yes

Reviewer #3: No

4. Have the authors made all data underlying the findings in their manuscript fully available (please refer to the Data Availability Statement at the start of the manuscript PDF file)?

Reviewer #2: Yes

Reviewer #3: No

5. Is the manuscript presented in an intelligible fashion and written in standard English?

Reviewer #2: Yes

Reviewer #3: Yes

6. Review Comments to the Author

Reviewer #2: Clinical readiness for essential maternal and child health services in Kenya: A cross-sectional survey

Thank you for the opportunity to review this paper

• The manuscript covers a significant health topic

• The quality of manuscript is good

• It meets the minimum requirements for scientific writing

Recommendation

• The manuscript to be considered for publication after authors’ attention to the review comments highlighted to the satisfaction of journal editor

Specific comments for authors’ consideration

Line 47… substitute the word “supply” with the word …” supplies” ……

Line 48, reorganize the sentence to read…” decrease pregnancy and postpartum complications”.

In the introduction line 53, authors should give teen pregnancies more weight as an independent variable besides population aging and increase in NCDs as a determinant of need for emergency obstetric care in LMIC. Authors to review the first sentence to read; Teen pregnancies, aging population and increase in non-communicable diseases………. delete the sentence on teen pregnancy in line 55 & 56

Line 70. Replace the word “these goals” with “the goals” …...

In Line 74, specify the providers to read “health care providers”

Line 124 & 125 …participants were excluded if they could not fill the survey questionnaire...; while line 126… reads of verbally administered interviews by study person; Authors to clarify the actual data collection tool adopted and procedure of data collection

Line 151…. and line 153… replace ….” supply” with “supplies”

Line 165…. The word “Supply” should be replaced with the… “type of supplies or category of supplies”. The author should replace the action verb supply with supplies throughout the document. Line 216, Table 3 column 1, row 1; line 220, 226, 231, 247, 265, 342, 347…. etc

Line 250. Qualify…...’providers’ as “health care providers”

Line 251. Add the article ‘the’ before providers to read... “the providers”

Line 275 to 277 carries a biased observation that adversely depicts the nurses as poor performers in post abortion when indeed the sampling was skewed and finding is not statistically significant. Authors to review the statement…...

Line 286… specify providers to read … “health care providers” ….

Line 286… substitute the word …” decreasing” ... with... “reducing”

Line 315, add the word ‘only’…..the statement should read partly .. legalized abortion only……..to specify when the law allows procurement of abortion

Line 317… replace the word …” decreasing” … with “…...reducing”

Line 356... review the statement that reads ...” supply availability” to read ….” supplies availability” or “availability of supplies”

References No. 30 & 31 have not been cited in the text… Authors to drop them or cite where applicable

Reviewer #3: Dear author,

The paper is insightful and introduces a new area of focus, however it needs some work on clarity as well as further improvements.

1. It is not clear what the objective of the paper is. It is not clear if the paper seeks to determine clinical readiness or just simply seeks to understand MCH care provision in the three counties. This has not been presented in a clear manner. From the title the focus is to determine clinical readiness among health care providers (line 99) but how this is done or has been done does not clearly come out in the paper. In line 102, the approach is stated but there is a lack of how clinical knowledge, training, and supply availability determine clinical readiness or how they link up.

2. Similarly, from my understanding, there is no outcome measure for clinical readiness (lines 150 -159), which begs the question what are the independent and dependent variables in this analysis and is there a framework that guides the study ? If so, it will be better to include it in the narrative in order to give direction or tie up all parts of the study in a more coherent manner.

3. In line 136-137 there is a contradiction, on one hand there is the claim that the questionnaires were validated by experts and then in the line that follows you indicate that no formal validation process was followed. This does not make sense, either validation was done or not but not both. By design, if a questionnaire has been applied and tested multiple times then there is some form of intrinsic validity in their use whether all or some of the questions from the questionnaire are used in the study.

4. Even though assessing clinical knowledge, training and resources independently provides valuable insights the study would be better if all this can be linked up to clinical readiness at the facility or individual level. As indicated in the paper the study was exploratory in nature and thus a simple linear model or a likelihood model would have sufficed in linking clinical readiness to the exploratory variables . The sample size (n=132) is sufficient to ensure convergence even if the study was to rely on a maximum likelihood estimation approach of overall clinical readiness either at the facility level or individual level. Also, the importance of comparing the correct questions between MCH topics with post abortion care chosen as the reference has not been clearly elucidated and its not clear why this was done.

5. Linear models with fixed effects require evidence of non-independence, heterogeneity and non-linearity within the data which has not been presented or alluded to within the paper. It also assumes that the explanatory variables are linearly related to the response. If there is evidence for any of this, then this should be included and clarified presented as a justification.

6. The statistical methodology applied is disjointed and fails to clearly address the objectives of the paper and thus the paper is not replicable in the same context elsewhere. It also makes it difficult to determine if the conclusions have been arrived at in an objective manner from the findings. From the paper its alluded that clinical readiness outcome might not be a continuous variable but dichotomous in nature. Considerations should therefore be made to develop a statistical model that links clinical readiness to training, resources, and clinical knowledge as that would make for a better paper.

7. PLOS authors have the option to publish the peer review history of their article (what does this mean?). If published, this will include your full peer review and any attached files.

**Do you want your identity to be public for this peer review?** For information about this choice, including consent withdrawal, please see our Privacy Policy.

Reviewer #2: **Yes: **DR. Angeline Chepchirchir Kirui

Reviewer #3: No

---

## [Decision Letter · Decision Letter 2]

15 Nov 2023

Clinical readiness for essential maternal and child health services in Kenya: A cross-sectional survey

PGPH-D-22-01510R2

Dear Dr Hagey,

We are pleased to inform you that your manuscript 'Clinical readiness for essential maternal and child health services in Kenya: A cross-sectional survey' has been provisionally accepted for publication in PLOS Global Public Health.

Best regards,

Adriana A. E. Biney

Academic Editor

**Academic Editor's Comments**

As you finalize the manuscript for publication, I suggest you include a bit more information in the Introduction to provide some context on MCH services in Kenya. Currently, the only specific reference to Kenya and access to MCH supplies is on page 6, line 77.  

In addition, please address the following:

- On page 22, line 295, include in brackets a statement that the table is not show (or you can refer to the table if it is indicated in the manuscript)

- On page 22, line 297, the male providers' average score was 8.9%. Please provide some explanation for the additional numbers (66.1% versus 57.1%) as this is not too clear. Is it compared to the total and females?

Reviewer Comments (if any, and for reference):

Reviewer's Responses to Questions

**Comments to the Author**

1. If the authors have adequately addressed your comments raised in a previous round of review and you feel that this manuscript is now acceptable for publication, you may indicate that here to bypass the “Comments to the Author” section, enter your conflict of interest statement in the “Confidential to Editor” section, and submit your "Accept" recommendation.

Reviewer #2: All comments have been addressed

Reviewer #3: All comments have been addressed

2. Does this manuscript meet PLOS Global Public Health’s publication criteria? Is the manuscript technically sound, and do the data support the conclusions? The manuscript must describe methodologically and ethically rigorous research with conclusions that are appropriately drawn based on the data presented.

Reviewer #2: Yes

Reviewer #3: Yes

3. Has the statistical analysis been performed appropriately and rigorously?

Reviewer #2: Yes

Reviewer #3: Yes

4. Have the authors made all data underlying the findings in their manuscript fully available (please refer to the Data Availability Statement at the start of the manuscript PDF file)?

Reviewer #2: Yes

Reviewer #3: Yes

5. Is the manuscript presented in an intelligible fashion and written in standard English?

Reviewer #2: Yes

Reviewer #3: Yes

6. Review Comments to the Author

Reviewer #2: The authors have adequately addressed shared comments raised in the first review

The authors have provided access to the primary data

The manuscript has been written using standard English language

The statistics applied in data analysis are appropriate

Reviewer #3: All my comments and suggestions have been addressed accordingly and now the paper offers the required clarity needed.

No other issues or comments but a minor adjustment on line 260 : Insert "correct" before questions.

7. PLOS authors have the option to publish the peer review history of their article (what does this mean?). If published, this will include your full peer review and any attached files.

**Do you want your identity to be public for this peer review?** For information about this choice, including consent withdrawal, please see our Privacy Policy.

Reviewer #2: No

Reviewer #3: No
